# What Is the Effect of Attributing Disordered Eating Behaviours to Food Addiction Versus Binge Eating Disorder? An Experimental Study Comparing the Impact on Weight-Based and Mental Illness Stigma

**DOI:** 10.3390/nu17132217

**Published:** 2025-07-04

**Authors:** Megan G. Molnar, Lindsey A. Snaychuk, Stephanie E. Cassin

**Affiliations:** Department of Psychology, Toronto Metropolitan University, Toronto, ON M5B 2K3, Canada; megan.molnar@torontomu.ca (M.G.M.); lindsey.snaychuk@torontomu.ca (L.A.S.)

**Keywords:** food addiction, binge eating disorder, diagnosis, stigma, weight bias

## Abstract

**Background/Objectives:** Food addiction (FA) and binge eating disorder share many overlapping features. Many individuals with binge eating disorder experience stigma; however, less is known about the stigma associated with food addiction. The current study examined the weight-based stigma and mental illness stigma associated with attributing disordered eating behaviours to an FA diagnosis or binge eating disorder diagnosis. **Methods:** Undergraduate students (*N* = 177) were randomly assigned to read one of three vignettes (FA, binge eating disorder, or control), all of which described a character experiencing the overlapping features of FA and binge eating disorder; the vignettes differed only regarding the diagnosis to which the eating behaviours were attributed. Participants then completed questionnaires assessing their attitudes towards mental illness and obesity followed by questionnaires assessing their own eating behaviours. **Results:** There were no significant between-group differences in mental illness stigma or weight-based stigma. Significant differences in stigma were found based on the perceived gender of the vignette character and participants’ own FA and binge eating disorder symptoms. **Conclusions:** Stigma may not differ based on the diagnosis ascribed to addictive-like eating behaviours. Women may be more stigmatized for addictive-like eating behaviours, and individuals who experience addictive-like eating may be more stigmatizing towards others with these behaviours.

## 1. Introduction

The concept of food addiction (FA) proposes that some individuals may experience an addiction-like response to certain foods, particularly ultra-processed hyperpalatable foods [1], which may prompt weight-promoting eating behaviours. Although the clinical features of FA have been investigated for decades and researchers have developed tools to assess FA [1,2,3], it is not an official diagnosis in the Diagnostic and Statistical Manual of Mental Disorders (DSM-5) [4].

Currently, there is debate over the conceptualization of addictive-like eating; specifically, whether it is best classified as a behavioural addiction or substance use disorder [5]. Highly processed foods with added fat or refined carbohydrates are most commonly associated with addictive-like eating behaviours [6], presumably because hyperpalatable foods activate the dopaminergic pathway which regulates reward sensitivity, impulse control, stress reactivity, and other factors involved in motivation [7]. There is evidence suggesting similar dysfunctional patterns of reward-related neural activation in addictive-like eating behaviour and substance dependence [8]. Most advocates for the behavioural conceptualization of FA conclude that the validity of FA from neurobiological evidence and rodent models, which affirm that food activates the brain’s reward system, is inadequate and there is insufficient evidence linking any common ingredient, combination of ingredients, or micronutrient as addictive [5,9]. Identifying the addictive component of the substance is necessary for the conceptualization of substance use disorders; however, identifying the specific component that makes food addictive is challenging because food typically contains many ingredients.

There is also debate regarding whether FA is sufficiently distinct from binge eating disorder (BED) to warrant a new diagnosis. There are many overlapping features between BED and FA, such as initiating food consumption in the absence of hunger, overindulging past the point of fullness, attempting to quit maladaptive eating behaviours but failing to do so, and continuing despite adverse effects [10,11,12]. Both BED and addictive-like eating behaviours are associated with high psychological distress [13] and eating behaviours that increase vulnerability for excess weight and obesity [11,14,15].

Despite some similarities, there are also notable differences between FA and BED. For example, some forms of maladaptive eating other than binge eating, such as grazing, are more characteristic of FA than BED [16]. In addition, there are some unique mechanisms underlying each condition, such as substance tolerance and withdrawal symptoms in FA that are not seen in those with BED [17]. Evidence for the co-occurrence of FA and BED also validates the construct of FA. Some individuals with BED report symptoms of FA whereas others do not, and those experiencing co-occurring FA and BED often endorse higher levels of depression, anxiety, and impulsivity than those who have either BED or FA [13,18,19].

In addition to considering the validity and clinical utility of diagnoses, it is important to take ethical considerations into account when generating new diagnoses, including the potential impact on stigma [20]. Stigma refers to negative attitudes towards a particular group, and stereotyping beliefs towards an individual who is a part of that group, which may result in discriminatory behaviours [21]. Many individuals with BED experience stigma, and those with BED and obesity may be at risk of “double” stigma due to the potential additive effect of weight-based stigma plus mental illness stigma [22], though evidence is mixed [23]. Studies suggest that BED is perceived as a problem of weakness or laziness [23] and low self-esteem or depression [24]. BED is also attributed to a failure of self-discipline, and individuals with BED are thought to have greater personal responsibility for their behaviour than individuals with other conditions [25]. Therefore, attributing BED to weakness or personal failure may result in more self-stigma in those with BED. Based on the aforementioned research, it seems that a BED diagnosis may not decrease weight-based stigma towards individuals with BED and obesity.

Efforts have been made to examine the weight-based stigma associated with FA given that, like individuals with BED, individuals with FA are at risk of experiencing both weight-based and mental illness stigma. Studies suggest that an FA “diagnosis” may increase both externalized stigma towards those with a higher weight [26,27] and internalized stigma [28]. Conversely, other studies have found that an FA “diagnosis” can actually have the opposite effect on stigma, such that individuals with FA are seen as having less personal control over their eating behaviours and being less at fault for their weight [29,30]. There is also evidence to suggest that externalized stigma does not differ between an FA or eating addiction “diagnosis” [28].

Overall, the limited research conducted to date on the stigma associated with FA has generated mixed results, which may be due, in part, to the different participant samples (e.g., university students [26,27,30], adults recruited through social networking sites [27,29], or adults recruited through crowdsourcing such as Mechanical Turk [28]) and the varied methodologies across studies. For example, participants in some studies read general informational passages attributing weight to FA or other factors (e.g., poor dietary choices or inactivity) [28,30], whereas others read vignettes or newspaper articles describing a specific character [26,27,29]. In the case of specific characters, some studies described the character’s weight [27,29] or included a picture of the character [26]. Finally, the studies included a variety of measures examining different facets of stigma.

### 1.1. Study Objectives

Many studies have compared stigma between several disorders and/or addictions [22,31,32,33]. Consistent with attribution theory [34], in comparison with individuals with physical-based stigmas (e.g., cancer, blindness), those with mental–behavioural stigmas (e.g., obesity, substance abuse) are judged as being more responsible for their condition because the condition is perceived as being within their personal control, and they elicit more negative affective reactions (e.g., greater blame and anger, less liking and pity) [33]. However, these attributions are potentially modifiable in response to information regarding causal factors outside of a person’s control that are implicated in the onset of the condition [33].

Surprisingly, no studies to our knowledge have compared stigma between FA and BED despite their substantial symptom overlap. Therefore, the objective of the present experimental study was to examine whether attributing disordered eating behaviours to FA or BED has implications for weight-based or mental illness stigma. This is an important empirical question because it separates the stigma of the diagnostic label from the stigma of the prominent behavioural features shared by both conditions, such as a lack of control over eating. All participants read a vignette describing a gender-neutral character (“JC”) experiencing the overlapping symptoms of FA and BED, and the character’s eating behaviours were attributed to either FA or BED (experimental conditions) or no explanation for eating behaviours was provided (control condition).

### 1.2. Hypotheses

It was hypothesized that both experimental conditions (FA vignette; BED vignette) would differ significantly from the control condition in mental illness stigma towards the vignette character (H_1_). It was hypothesized that both experimental conditions (FA vignette; BED vignette) would differ significantly from the control condition in weight-based stigma towards the vignette character (H_2_). Given that previous research has generated mixed results regarding whether an FA label increases or decreases stigma, the hypotheses were non-directional.

### 1.3. Exploratory Analyses

Several exploratory analyses were also conducted. To our knowledge, no previous research has examined how gender impacts FA stigma. To fill this gap, we compared stigma in participants who assumed the vignette character was a man versus participants who assumed the vignette character was a woman. In addition, we explored whether participants’ personal experience of FA and binge eating symptoms impacted their attitudes towards the vignette character.

## 2. Methods

### 2.1. Participants

Undergraduate students (*N* = 282) enrolled in introductory psychology courses at Toronto Metropolitan University (TMU) were recruited to participate in this study. There were no exclusion criteria for this study.

### 2.2. Procedure

All procedures were approved by the Toronto Metropolitan University Research Ethics Board (REB#: 2024-374). Participants completed the study online through the online survey platform Qualtrics. The consent form informed participants that they would be asked to read a description of an individual and complete a series of questionnaires about their attitudes towards mental illness and weight. The consent form did not disclose the true purpose of the study (i.e., to examine whether attributing disordered eating behaviours to FA or BED has implications for weight-based or mental illness stigma) to avoid biasing responses. Participants who provided consent were randomly assigned to read one of the three vignettes: FA, BED, or a no diagnosis control condition. The vignettes all described a character experiencing the overlapping symptoms of FA/BED. The vignette character had a gender-neutral name (“JC”) to reduce the potential impact of gender on weight-based or mental illness stigma. Depending on which condition participants were assigned to, the symptoms described in the vignette were explicitly attributed to the following: (1) FA, (2) BED, or (3) not attributed to a diagnosis (control condition). The vignettes (described below) were identical in all other respects:

“JC is a first-year university student (experiencing food addiction OR experiencing binge eating disorder OR no diagnostic label provided [control]). During the past few months, JC has felt driven to engage in weight-promoting eating behaviours, such as eating large amounts of food in a short amount of time and compulsive overeating. JC has started to crave and consume more food with added fat and refined carbohydrates, such as cookies, ice cream, chips, and fast food. In particular, JC tends to consume a lot of these foods when they are experiencing stressors, negative emotions, or negative feelings about food. JC experiences a lack of control over eating (like being on auto-pilot) and ends up eating much more food than intended. Despite eating to the point of feeling sluggish or sick, JC continues to overeat. Overeating also causes JC to spend a lot of time experiencing negative emotions which prevents them from working, spending time with friends and family or engaging in other important activities or recreational activities that JC enjoys.”

After reading the vignette, participants were asked to indicate the gender of the individual described in the vignette to explore whether they made any assumptions given that gender was intentionally not described. Participants then completed several questionnaires (described below) assessing attitudes towards mental illness and people living in larger bodies. Participants were then asked to report on their own FA and binge eating symptoms to assess whether their own eating behaviours were associated with weight-based and mental illness stigma. Finally, participants completed a demographics survey, which also asked them to indicate their height and weight in order to calculate their body mass index (BMI).

### 2.3. Measures

#### 2.3.1. Mental Illness Stigma

The Mental Illness Stigma Scale (MISS) [35] is a 28-item Likert-style measure that includes seven subscales (anxiety, relationship disruption, poor hygiene, visibility, treatability, professional efficacy, and recovery) assessing stigmatizing attitudes towards people with mental illness [35]. The scale was adapted to specifically refer to the character described in the vignette (“JC”). Participants were asked to indicate the extent to which they agree (1 = completely disagree; 7 = completely agree) with each statement (i.e., “It would be difficult to have a close meaningful relationship with JC”). Total scores on the MISS can range from 28 to 196, with higher scores indicating greater stigma towards those with mental illness.

The Affective Reactions Scale [36] assesses emotional reactions towards a target. Participants were asked to rate the extent to which they feel certain emotions towards the character described in the vignette (“JC”) on a 6-point scale (1 = not at all; 6 = very much). The scale assesses negative affective reactions (anger, irritation, disgust, annoyance, dislike) and sympathetic affective reactions (concern, sympathy, pity). Scores range from 5 to 30 for the negative affective reactions subscale, with higher scores indicating greater negative reactions towards JC. Scores range from 3 to 18 for the sympathetic affective reactions subscale, with higher scores indicating greater sympathetic reactions towards JC.

#### 2.3.2. Weight-Based Stigma

The Anti-fat Attitudes Questionnaire (AFA) [37] is a 13-item Likert scale that assesses anti-fat attitudes. Participants were asked to rate the extent to which they agree with each statement on a Likert-style scale (0 = very strongly disagree; 9 = very strongly agree). The questionnaire includes 3 subscales: dislike towards fat people (i.e., “I really don’t like fat people much”), willpower or controllability regarding weight (i.e., “Some people are fat because they have no willpower”), and fear of fat (i.e., “I worry about becoming fat”). The fear of fat subscale was not included in the current study because it assesses stigma directed towards the self rather than others. Total scores (for the 2 subscales included in the current study) range from 0 to 90, with higher scores indicating greater weight-based stigma.

The Fat Phobia Scale–short form [38] is a 14-item questionnaire with a 1–5 Likert-style scale that assesses beliefs and feelings towards individuals who are fat. Respondents are shown two contrasting adjectives (e.g., “attractive, unattractive”; “insecure, secure”) and are asked to place an ‘X’ on the line closest to the adjective they believe best describes people who are fat. Total scores range from 1 to 5, with higher scores indicating greater weight-based stigma.

The Universal Measure of Bias-Fat Questionnaire (UMB-FAT) [39] is a 20-item scale designed to assess general attitudes towards people who are fat. Participants are asked to rate the extent to which they agree with various statements on a Likert-style scale (1 = strongly agree; 7 = strongly disagree). The questionnaire includes 4 subscales: adverse judgment (i.e., “Fat people have bad hygiene”), social distance (i.e., “I don’t enjoy having a conversation with a fat person”), attraction (i.e., “I find fat people to be sexy”), and equal rights (i.e., “Special effort should be taken to make sure that fat people have the same salaries as other people”). Each subscale can range from 5 to 35, with higher scores indicating greater weight-based stigma.

#### 2.3.3. Participant Eating Behaviours

The Binge Eating Scale (BES) [40] is a 16-item questionnaire designed to assess the affective, behavioural, and cognitive aspects of binge eating. Participants are shown items with multiple statements about behaviours, attitudes, and thoughts regarding eating (i.e., “I have days when I can’t seem to think about anything but food”) and are asked to choose the statement that best describes them. Scores range from 0 to 46, with higher scores indicating greater binge eating symptomatology. The scale classifies individuals into three severity categories: no/low binge eating (scores of 17 or lower), moderate binge eating (scores of 18 to 26), and severe binge eating (scores of 27 or higher).

The Modified Yale Food Addiction Scale 2.0 (mYFAS 2.0) [41] is a 13-item questionnaire designed to assess FA symptoms by using adapted substance use disorder criteria from the DSM-5. Participants are asked how often in the past 12 months they have experienced certain FA symptoms as well as distress and impairment (i.e., “My eating behaviour caused me a lot of distress”; “My friends or family were worried about how much I overate”) on a Likert scale (0 = never; 7 = everyday). Higher scores on the scale indicate greater FA symptomatology. Although the mYFAS 2.0 is not a diagnostic tool, the severity of FA can be scored according to the number of criteria endorsed: no FA (fewer than 2 symptoms and/or no impairment/distress), mild (2–3 symptoms plus impairment or distress), moderate (4–5 symptoms plus impairment or distress), severe (6 or more symptoms plus impairment or distress). In the current study, this scale was scored based on the severity of FA and the clinical significance criteria. Those who scored in the mild, moderate, or severe categories were considered to exceed the cutoff for probable FA.

### 2.4. Data Analysis

Data analysis was conducted using IBM SPSS Version 27. Participants who skipped or provided the same response on the majority of questionnaire items and those who did not re-consent to having their data analyzed following debriefing had their data removed. The mean duration for participants (excluding outliers) to complete this study was 36 min. Participants who completed the study in less than 12 min did not have their data included in the data analysis because it was deemed too fast to read through the vignette and questionnaires. After data cleaning, the final sample consisted of 177 participants (FA condition *n* = 61; BED condition *n* = 52; control condition *n* = 64).

One-way ANOVAs were conducted to examine group differences in stigma scores between different vignette conditions (FA, BED, control) and between participants with different levels of binge eating severity according to the BES, whereas Kruskal–Wallis H tests were used for dependent variables that violated the assumption of normality. Independent samples *t*-tests were used to compare stigma scores between participants who perceived the vignette character as being a man versus those who perceived the vignette character as being a woman, and to compare stigma scores between participants who exceeded the cutoff for probable FA to those who did not according to the mYFAS 2.0. Mann–Whitney U tests were used for dependent variables that violated the assumption of normality.

## 3. Results

### 3.1. Participant Characteristics

Participants (*N* = 177) had a mean age of 19.5 years (SD = 2.37). They were primarily women (88.1%) and were ethnically diverse (25.4% South Asian; 16.4% White). A sizable minority of participants exceeded the cutoff for probable FA (18.3%) or for severe binge eating (12.4%). Participant demographics and clinical characteristics are presented in Table 1.

### 3.2. The Effect of the Vignette Character’s Diagnostic Label on Stigma

The mean scores on mental illness stigma measures are presented in Table 2. There were no significant group differences in mental illness stigma towards the vignette character (regardless of their diagnostic label) on any of the Mental Illness Stigma subscales (treatability, relationship disruption, professional efficacy, recovery, visibility, anxiety, and hygiene). There were also no significant group differences on either of the Affective Reactions subscales (negative affect or sympathetic affect).

The mean scores on weight-based stigma measures are presented in Table 3. There were no significant group differences in weight-based stigma (regardless of the vignette character’s diagnostic label) on the Fat Phobia Scale, on either subscale of the Anti-fat Attitudes Questionnaire (willpower or dislike), or on any subscales of the Universal Measure of Bias-Fat Questionnaire (social distance, attraction, equal rights, and adverse judgement). Collectively, these results indicate that the diagnostic label that the eating behaviours were attributed to did not significantly impact weight-based or mental illness stigma.

### 3.3. The Effect of the Assumed Gender of the Vignette Character on Stigma

When prompted to identify the gender of the vignette character (“JC”), most participants identified the gender as being ‘not specified’ (*n* = 83) followed by a man (*n* = 68) and a woman (*n* = 26). There were significant differences on several subscales of the MISS based on whether participants assumed the vignette character was a man versus a woman (see Table 4). Participants who assumed the character was a man reported significantly higher treatability, professional efficacy, and recovery scores compared with those who assumed the character was a woman. In contrast, participants who assumed the vignette character was a woman reported significantly higher relationship disruption scores compared with participants who assumed the vignette character was a man. There were also significant differences on several of the UMB-FAT subscales. Participants who assumed the vignette character was a woman reported significantly higher adverse judgement scores and social distance scores compared with participants who assumed the vignette character was a man.

### 3.4. The Effect of Participant FA Score on Stigma

Mann–Whitney U analysis indicated significant differences in stigma scores towards the vignette character based on whether participants exceeded the mYFAS 2.0 cutoff score for probable FA (*U* = 1615.0, *p* = 0.008). There was a significant difference in recovery attitudes (*U* = 1615.0, *p* = 0.008). The mean rank for participants with probable FA was higher than those without FA (92.7 vs. 66.9). There was also a significant difference in sympathetic affect (*U* = 1779.0, *p* = 0.048). The mean rank for participants with probable FA was higher than those without FA (103.9 vs. 84.4). An independent sample *t*-test indicated a significant difference in willpower attitudes (*t*_(173)_ = −2.79, *p* = 0.003). Participants with probable FA had a higher mean score than those without FA (15.4 vs. 12.3). The group differences on other stigma subscales were not significant.

### 3.5. The Effect of Participant Binge Eating Score on Stigma

The Kruskal–Wallis H tests indicated significant differences in mental illness and weight-based stigma based on participants’ binge eating symptom severity according to the BES. The results indicated that there were statistically significant differences in recovery attitudes (*H*_(2)_ = 10.6, *p* = 0.005), sympathetic affect reactions (*H*_(2)_ = 7.8, *p* = 0.02), and willpower attitudes (*H*_(2)_ = 20.3, *p* = 0.049) towards the vignette character. Participants who exceeded the cutoff score for severe binge eating symptoms had a lower mean rank (*M* = 57.2) on recovery attitudes than participants with moderate (*M* = 88.2) or no/low (*M* = 94.7) binge eating symptoms. Participants who exceeded the cutoff for severe binge eating symptoms also had a higher mean rank (*M* = 115.6) on sympathetic affective reactions compared with those with moderate (*M* = 95.3) or no/low (*M* = 88.3) binge eating symptoms. Finally, participants who exceeded the cut off for severe binge eating symptoms had a higher mean rank (*M* = 111.5) for willpower attitudes than those with moderate (*M* = 95.3) or no/low (*M* = 83.7) binge eating symptoms. The group differences on other stigma subscales were not significant.

## 4. Discussion

This study examined whether attributing disordered eating behaviours to a “diagnosis” of FA or BED has implications for weight-based or mental illness stigma. According to attribution theory, stigma may be reduced by providing a causal explanation for a behaviour or condition that reduces the perception of personal responsibility [33]. The primary hypotheses were unsupported, as there were no significant group differences in stigma regardless of the “diagnosis” to which the vignette character’s eating behaviours were attributed. The findings of the current study suggest that merely attributing loss-of-control eating behaviours and body weight to a diagnostic label of FA or BED may not provide a sufficient causal explanation to alter perceptions of controllability and personal responsibility for eating behaviours. Guided by attribution theory, it may be more impactful to provide education regarding the complex interplay of factors that impact eating behaviours and body weight (including genetics, biological factors, and the ubiquitousness of ultra-processed foods that are engineered to be hyperpalatable), information that is not conveyed through a diagnostic label alone.

The lack of significant differences in stigma regardless of the vignette character’s “diagnosis” suggests that participants did not stigmatize the character more or less based on the condition ascribed to their eating behaviours. These findings align with a previous study that found no differences in stigma depending on whether obesity was attributed to FA, poor lifestyle factors, or no explanation provided for obesity causal factors [30]. In addition, weight-based and mental illness stigma scores were relatively low across all measures in our sample. Considering the potential ethical implications of formulating a diagnosis of FA [20], the findings from the current study suggest that an FA “diagnosis” may not necessarily increase weight-based or mental illness stigma towards those with addictive-like eating behaviours compared with a BED diagnosis. However, an FA “diagnosis” was also not stigmatized significantly less than a BED diagnosis in our study, suggesting that an FA diagnosis may not be any better at alleviating stigma than a BED diagnosis.

Exploratory analyses indicated differences in stigma towards the vignette character based on the assumed gender (i.e., man or woman) of the vignette character. To our knowledge, this study is the first to explore whether stigma regarding addictive-like eating behaviours differs depending on the gender of the vignette character. Interestingly, although the vignette character was intentionally given a gender-neutral name (“JC”), more participants assumed that the character was a man. The results suggested that participants who assumed the vignette character was a man reported the vignette character as being more likely to have their condition adequately recognized, to receive satisfactory treatment for their condition, and to recover compared with participants who assumed the character was a woman.

In the treatment and recovery contexts of pain, studies suggest that men may be seen as being stoic and in control, whereas women are seen as being more sensitive to pain, resulting in them being perceived as “weaker” [42]. Moreover, women’s symptoms are often seen as being invisible and/or psychological by health care providers [42]. Although it is not clear whether addictive-like eating behaviours are perceived as being more common among men or women, in general, women experience more dismissal of their health-related concerns by healthcare providers than men [43,44] which may affect treatment, professional efficacy, and recovery outcomes.

Participants who assumed the vignette character was a woman reported higher scores of relationship disruption compared with participants who assumed the vignette character was a man, suggesting the belief that being in a relationship with the vignette character would be too demanding and too emotional. This finding aligns with studies on pain suggesting that women are perceived as being more “dramatic” about their pain than men [45]. Considering the importance of social support in recovery, stress relief, and general wellbeing [46,47], relationship disruption attitudes that may deter individuals from forming or maintaining relationships with individuals with addictive-like eating behaviours can be harmful. Additionally, participants who assumed the vignette character was a woman reported higher adverse judgement scores compared with participants who assumed the vignette character was a man. Findings comparing weight-based stigma associated with gender are mixed, with some studies suggesting that men experience weight-based stigma just as much as women [48,49] and others suggesting that women experience more weight-based stigma [50]. However, women tend to internalize weight-based stigma more so than men [48]. The internalization of weight bias can be related to maladaptive coping responses and psychological distress [51]. Future research should investigate gender differences in weight-based stigma towards individuals with addictive-like eating behaviours further.

With regard to the relationship between participants’ own eating behaviours and stigma towards the vignette character and weight-based stigma in general, the findings suggested that participants who did not meet the cutoff score for FA reported that the vignette character was more likely to recover from their condition than participants who exceeded the cut off score. Therefore, it is possible that participants with personal experience of FA symptoms may have projected their own attitudes onto the vignette character which may reflect their own struggles to cope with and recover from disordered eating behaviours. The severity of FA has been found to be related to anti-fat attitudes and low eating self-efficacy [52], which suggests that individuals with probable FA may have low self-efficacy in recovery attitudes regarding their own eating behaviours. Participants who experienced probable FA endorsed stronger beliefs that obesity is due to a lack of willpower. Our findings contrast with a prior study that found no significant differences in externalized weight-based stigma towards a vignette character with obesity who either had FA or did not have FA, and no significant differences in stigma based on participants’ own perceived FA [28]. Importantly, the measure (AFA) used in the current study assessed anti-fat stigma in general rather than anti-fat stigma towards the vignette character. Therefore, it is possible that participants projected their own internalized weight-based stigma onto the vignette character.

Although participants who exceeded the cutoff for probable FA reported more stigmatizing attitudes on the aforementioned measures, they also reported more sympathetic affective reactions towards the vignette character than participants who did not meet the cutoff. This finding was also true of participants exceeding the cutoff for binge eating symptoms. Individuals with personal experience of FA and BED symptoms may relate more to the vignette character than those without FA and therefore may be more likely to empathize with the vignette character. This finding aligns with previous research which found that individuals who experienced stigma towards their own condition (i.e., stuttering, obesity, and mental illness) show more empathy towards others with that same condition [53].

The present study also explored whether participants’ binge eating symptom severity impacted their attitudes towards the vignette character. Participants with severe binge eating symptoms were more likely to believe that obesity is due to a lack of willpower and less likely to believe that the vignette character would recover from their condition than participants with no/low binge eating severity. Participants experiencing current binge eating symptoms may have lower self-efficacy in recovery attitudes and may be projecting that attitude onto the vignette character. However, they may also internalize societal stereotypes that associate BED with laziness and lack of willpower [24,25,54] or may consider BED to be primarily a problem of low self-esteem or depression which could then impact one’s ability to recover sufficiently [23]. The similar findings between participants with FA symptoms and BED symptoms may be a reflection of the strong positive correlation between FA symptoms and BED symptoms [19], which was also evident in our sample.

### Limitations and Future Directions

The current study aimed to address knowledge gaps in the literature on the stigma associated with FA and BED. The strengths of this study include the experimental design, use of validated measures, and the gender-neutral vignette, which, to our knowledge, is novel in FA stigma research. There are several limitations that need to be considered when interpreting the results of this study. First, this study recruited undergraduate psychology students so the results of the current study may not generalize to a more diverse community sample. Second, a larger sample size would have been needed to detect a small effect so it may be premature to conclude that the diagnostic label ascribed to eating behaviours does not have a significant impact on stigma. Future research should recruit a large and diverse sample from the general population, ideally with enough participants in each demographic subgroup (i.e., gender, gender identity, ethnicity) to examine whether participant demographic characteristics impact stigma regarding FA or BED. Third, although the first sentence of the vignette, which indicated the condition with which the vignette character was diagnosed, was written in bold to increase the salience of the diagnostic label, this study could have been strengthened by including a manipulation check such as a multiple choice or open response item asking participants to report the condition (if any) with which the vignette character had been diagnosed. It would also be valuable to replicate and extend the study by experimentally manipulating the gender of the vignette character to examine the causal effect of patient gender on stigma. Finally, this study assessed stigmatizing attitudes towards a vignette character in an experimental context, and the extent to which these results extend to real people who experience disordered eating symptoms is currently unknown. Future research should continue to explore the stigma associated with FA and BED, as well as the role of patient factors (e.g., gender) and participant factors (e.g., eating behaviours) in mental illness stigma and weight-based stigma.

## 5. Conclusions

It is important to consider the validity, clinical utility, and potential ethical implications when formulating new diagnoses. The present study addressed gaps in the literature regarding the weight-based and mental illness stigma associated with FA. The results suggest that a “diagnosis” of FA may not result in more (or less) stigma than a diagnosis of BED or identical eating behaviours that are not labeled with a diagnosis. Although controversy still exists regarding the validity and clinical utility of FA, and whether the symptoms are sufficiently distinct from BED to warrant a separate diagnosis, the results of the current study, if replicated in a large community sample, suggest that the diagnostic label of FA is unlikely to impact stigma.

## Figures and Tables

**Table 1 nutrients-17-02217-t001:** Participant Demographic Characteristics and Clinical Characteristics.

		Control (*n* = 64)	FA (*n* = 61)	BED (*n* = 52)
		*n*(%)	*n*(%)	*n*(%)
**Demographic Characteristics**				
Sex				
	Male	10(15.6)	7(11.5)	4(7.7)
	Female	54(84.4)	54(885)	48(92.3)
Gender				
	Man (Cis)	9(14.1)	7(11.5)	4(7.8)
	Woman (Cis)	51(79.7)	51(83.6)	29(90.2)
	Man (Trans)	1(1.6)	1(1.6)	2(2.0)
	Non-binary	1(1.6)	2(3.3)	0(0.0)
	Prefer not to specify	2(3.1)	0(0.0)	0(0.0)
Ethnicity				
	Asian—East	5(7.8)	4(6.6)	2(3.8)
	Asian—South	18(28.1)	12(19.7)	15(28.8)
	Asian—South East	10(15.6)	5(8.2)	7(13.5)
	Black—African	2(3.1)	6(9.8)	2(3.8)
	Black—Caribbean	4(6.3)	5(8.2)	4(7.7)
	First Nations	1(1.6)	0(0.0)	0(0.0)
	Latin American	2(3.1)	0(0.0)	0(0.0)
	Middle Eastern	6(9.4)	3(4.9)	7(13.5)
	White—European	7(10.9)	13(21.3)	9(17.3)
	White—North American	4(6.3)	6(9.8)	4(7.7)
	Mixed heritage	5(7.8)	5(0.9)	4(7.7)
	Prefer not to specify	0(0.0)	2(3.3)	0(0.0)
**Clinical Characteristics**				
Food Addiction Severity Category				
	None	50(79.4)	47(77.0)	46(90.2)
	Mild	3(4.8)	3(4.9)	1(2.0)
	Moderate	4(6.3)	0(0.0)	1(2.0)
	Severe	6(9.5)	11(18.0)	3(5.9)
Binge Eating Severity Category				
	No/Low	46(72.0)	41(67.2)	40(76.9)
	Moderate	8(12.5)	12(19.7)	8(15.4)
	Severe	10(15.6)	8(13.1)	4(7.7)
BMI Category				
	>18.5	8(14.0)	9(15.8)	8(16.0)
	18.5–24.9	31(54.4)	33(57.9)	29(58.0)
	25.0–29.9	11(19.3)	12(21.1)	9(18.0)
	<30.0	7(12.3)	3(5.4)	4(8.0)

Note. FA = Food Addiction; BED = Binge Eating Disorder. FA symptoms were assessed using the modified Yale Food Addiction Scale 2.0. Binge eating symptoms were assessed using the Binge Eating Scale. BMI = body mass index (kg/m^2^).

**Table 2 nutrients-17-02217-t002:** Scores on measures of mental illness stigma as a function of vignette condition.

Measure	Control (*n* = 64)	FA (*n* = 61)	BED (*n* = 52)	Statistical Test	*p*
	*M*(SD)	*M*(SD)	*M*(SD)		
**Mental Illness Stigma Scale (MISS)**					
Treatability	13.1(2.2)	13.2(2.7)	13.1(1.8)	*H* = 0.77	0.68
Anxiety	18.0(5.7)	18.0(6.3)	17.0(5.4)	*F* = 0.57	0.57
Hygiene	11.1(4.3)	11.5(4.1)	11.5(3.9)	*F* = 0.18	0.84
Relationship disruption	15.7(5.1)	16.7(5.2)	15.40(4.9)	*H* = 2.0	0.35
Professional efficacy	8.4(2.3)	8.9(2.8)	9.0(1.9)	*H* = 2.4	0.29
Recovery	10.0(1.6)	9.8(2.1)	10.2(1.5)	*H* = 0.84	0.66
Visibility	13.3(2.9)	12.6(3.5)	12.5(3.0)	*H* = 3.3	0.19
**Affective Reactions (AR**)					
Negative affect	4.0(4.3)	3.6(4.7)	3.8(4.6)	*H* = 0.57	0.75
Sympathetic affect	11.2(3.5)	10.8(4.0)	11.7(3.1)	*H* = 1.4	0.50

Note. FA = Food Addiction; BED = Binge Eating Disorder. None of the group differences were statistically significant. ANOVAs (*F*) were conducted for variables with normally distributed data whereas Kruskal–Wallis H Tests (*H*) were conducted for variables with data that were not normally distributed.

**Table 3 nutrients-17-02217-t003:** Scores on measures of weight-based stigma as a function of vignette condition.

Measure	Control (*n* = 64)	FA (*n* = 61)	BED (*n* = 52)	Statistical Test	*p*
	*M*(SD)	*M*(SD)	*M*(SD)		
**Anti-fat Attitudes Questionnaire (AFA)**					
Dislike	9.9(11.2)	10.5(9.3)	10.5(9.5)	*H* = 1.2	0.54
Willpower	11.6(6.4)	14.0(5.7)	13.0(5.2)	*F* = 2.8	0.06
**Fat Phobia Scale (FBS)**					
Fat phobia	2.7(0.61)	2.6(0.5)	2.6(0.5)	*H* = 0.12	0.94
**Universal Measure of Bias-Fat (UMB-FAT) Questionnaire**					
Adverse judgement	14.7(5.9)	14.3(6.0)	15.1(5.4)	*H* = 0.53	0.77
Social distance	12.4(4.8)	13.3(4.9)	12.8(4.7)	*F* = 0.58	0.56
Attraction	19.3(7.0)	21.1(5.7)	20.4(5.4)	*H* = 2.2	0.33
Equal rights	11.0(5.0)	12.0(6.5)	11.3(5.1)	*H* = 0.50	0.78

Note. FA = Food Addiction; BED = Binge Eating Disorder. None of the group differences were statistically significant. ANOVAs (*F*) were conducted for variables with normally distributed data whereas Kruskal–Wallis H Tests (*H*) were conducted for variables with data that were not normally distributed.

**Table 4 nutrients-17-02217-t004:** Scores on measures of stigma as a function of assumed gender of the vignette character.

Measure	Man (*n* = 68)	Woman (*n* = 26)	Statistical Test	*p*
	*M*(SD)	*M*(SD)		
**Mental Illness Stigma Scale (MISS)**				
Treatability	13.5(1.7)	11.4(3.0)	*U* = 494.5	<0.001
Hygiene	11.7(3.9)	11.9(4.1)	*t* = −0.194	0.423
Anxiety	17.7(5.2)	19.5(5.6)	*t* = −1.44	0.077
Visibility	12.8(3.0)	12.7(3.3)	*t* = 0.219	0.414
Professional efficacy	8.9(2.2)	7.8(2.5)	*U* = 651.0	0.044
Recovery	10.1(1.5)	8.8(2.3)	*U* = 5.46	0.003
Relationship disruption	15.1(4.2)	17.7(4.9)	*t* = −2.54	0.006
**Affective Reactions (AR)**				
Negative affect	4.2(4.5)	5.2(5.3)	*t* = −0.947	0.173
Positive affect	11.1(3.4)	12.4(3.9)	*t* = −1.57	0.60
**Anti-fat Attitudes Questionnaire (AFA)**				
Dislike	9.2(9.5)	13.5(12.4)	*U* = 767.0	0.32
Willpower	13.2(6.0)	15.3(5.8)	*t* = −1.51	0.066
**Universal Measure of Bias-Fat (UMB-FAT) Questionnaire**				
Adverse judgement	14.8(5.3)	17.0(5.5)	*t* = −1.81	0.037
Attraction	20.1(6.0)	22.0(7.1)	*t* = −1.303	0.098
Equal rights	11.5(5.4)	12.5(6.9)	*U* = 849.0	0.77
Social distance	12.7(4.5)	14.8(5.6)	*t* = −1.88	0.031

Note: *t*-tests were conducted for variables with normally distributed data whereas Mann–Whitney U tests were conducted for variables with data that were not normally distributed.

## Data Availability

Available upon request from the corresponding author. The data are not publicly available due to ethical reasons.

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
