# Peer review of "What Is the Effect of Attributing Disordered Eating Behaviours to Food Addiction Versus Binge Eating Disorder? An Experimental Study Comparing the Impact on Weight-Based and Mental Illness Stigma"

_nutrients, 2025, doi:10.3390/nu17132217_

Round 1
Reviewer 1 Report
Comments and Suggestions for Authors
This is an interesting experimental study examining whether attributing disordered eating behaviors to food addiction versus binge eating disorder affects weight-based or mental illness stigma. The topic is interesting and the paper is well-written. I have several comments to improve the manuscript further:
1. First. I noticed that while the introduction appropriately highlights the overlap between FA and BED, the manuscript insufficiently distinguishes the conceptual and clinical significance of comparing them in terms of stigma. A clearer theoretical justification for why diagnostic labels might affect stigma is needed.
2. The review notes that stigma research for FA is mixed, yet it does not critically engage with why prior findings are inconsistent. This can be strengthen further
3. No power analysis was reported. Given the null findings, readers need assurance that the study was adequately powered to detect small-to-medium effects.
4. The null hypothesis significance testing approach is insufficiently informative in the context of non-significant findings. The authors should also run Bayesian or equivalence test to examine the null further
5. Only partial data on demographic breakdowns (e.g., BMI, ethnicity) are contextualized in interpretation. The role of participant BMI or racial background in stigma perceptions could be explored or at least acknowledged as a limitation.
6. There should be discussion about the lack of manipulation check. While the vignette was standardized across conditions except for diagnosis, the manipulation may have lacked sufficient salience.
7. I feel that the discussion misses an opportunity to explore why stigma may not differ by diagnostic label. Attribution theory or essentialism literature could be used to speculate further.
Author Response
Reviewer #1 comments:
Overall Comment: This is an interesting experimental study examining whether attributing disordered eating behaviours to food addiction versus binge eating disorder affects weight-based or mental illness stigma. The topic is interesting and the paper is well-written.
Response: We thank the reviewer for their positive feedback and appreciate their individual comments to help strengthen our manuscript.
Comment 1: I noticed that while the introduction appropriately highlights the overlap between FA and BED, the manuscript insufficiently distinguishes the conceptual and clinical significance of comparing them in terms of stigma. A clearer theoretical justification for why diagnostic labels might affect stigma is needed.
Response: We appreciate the suggestion to clarify the conceptual and clinical significance of comparing FA and BED with regard to stigma. We have added a section regarding attribution theory to the introduction (page 3, line 100) and we have clarified the objective of the study (page 3, line 109) as follows: “Therefore, the objective of the present experimental study was to examine whether attributing disordered eating behaviours to FA or BED has implications for weight-based or mental illness stigma. This is an important empirical question because it separates the stigma of the diagnostic label from the stigma of the prominent behavioural features shared by both conditions, such as lack of control over eating”.
Comment 2: 2. The review notes that stigma research for FA is mixed, yet it does not critically engage with why prior findings are inconsistent. This can be strengthened further.
Response: We did not include this information in our original submission in an attempt to keep the introduction succinct; however, we appreciate the reviewer’s encouragement to speculate about the mixed findings and we have added a section to the introduction to help contextualize these mixed findings (page 2, line 88).
Comment 3: No power analysis was reported. Given the null findings, readers need assurance that the study was adequately powered to detect small-to-medium effects.
Response: A power analysis indicated that a sample size of 1548 would be needed to detect a small effect; however, we aimed to power our study to detect a medium effect because a small effect in this context is unlikely to be clinically meaningful. A sample size of 252 participants would be required to detect a medium effect. We recruited 282 participants; however, the final sample consisted of 177 participants following data cleaning because we removed participants who provided data with questionable validity (i.e., those who completed the study too quickly to read through the vignettes and questionnaires, those who provided the same response on the majority of questionnaire items). Although the final sample included in data analysis was unfortunately smaller than we aimed for, we believe the decision to remove participants who provided data of questionable validity improves the quality of the data that was analyzed. It is also worth noting that with the exception of a group difference that was nearly significant (p = 0.06) on one subscale (Antifat Attitudes Questionnaire - Willpower subscale), none of the other group differences between diagnostic labels approached significance so it is unlikely a larger sample size would have impacted the pattern of results. We have included sample size and insufficient power as a study limitation (page 13, line 453).
Comment 4: The null hypothesis significance testing approach is insufficiently informative in the context of non-significant findings. The authors should also run Bayesian or equivalence test to examine the null further.
Response: Thank you for this thoughtful suggestion. It was hypothesized that both experimental conditions (FA vignette; BED vignette) would differ significantly from the control condition in mental illness stigma towards the vignette character (H1) and in weight-based stigma towards the vignette character (H2). Given that we hypothesized a significant difference between the experimental groups vs. the control group, we believe that testing for a statistically significant difference is the appropriate statistical test. Given the non-significant difference between the diagnostic labels, we do agree that it would be interesting to conduct an equivalence test to examine whether the group difference is small enough to be considered practically equivalent; however, we did not include this analysis in our manuscript because we did not hypothesize that the groups would be equivalent.
Comment 5: Only partial data on demographic breakdowns (e.g., BMI, ethnicity) are contextualized in interpretation. The role of participant BMI or racial background in stigma perceptions could be explored or at least acknowledged as a limitation.
Response: We collected participant BMI and demographic characteristics primarily for the purpose of describing the participant sample and potential limitations regarding generalizability of findings.We appreciate the reviewer’s feedback and agree it would be informative to further explore the impact of participant BMI and demographic characteristics on weight-based and mental health stigma. We chose to focus on participants’ personal experience with binge eating and food addiction (i.e., BES and mYFAS scores) because they were most relevant to the current study, and we were cautious of not conducting too many additional analyses given the relatively small sample size, particularly within each separate demographic category (i.e., there were insufficient numbers of participants with high BMI, diverse gender identities, or racial backgrounds aside from White or Asian to draw meaningful interpretations). We have included the following statement in the future research directions (page 13, line 456): “Future research should recruit a large and diverse sample from the general population, ideally with enough participants in each demographic subgroup (i.e., gender, gender identity, ethnicity) to examine participant demographic characteristics impact stigma regarding food addiction or binge eating disorder.”
Comment 6: There should be discussion about the lack of manipulation check. While the vignette was standardized across conditions except for diagnosis, the manipulation may have lacked sufficient salience.
Response: We thank the reviewer for this suggestion. To increase the salience of the diagnostic label, the first sentence of the vignette which indicated the condition that the vignette character was diagnosed with was bolded. We also excluded participants from data analysis who completed the study too quickly to actually read through the vignette. We agree that it would have been helpful to include a manipulation check such as a multiple choice or open response item asking participants to report the condition (if any) that the vignette character had been diagnosed with. We have included the lack of manipulation check as a study limitation (page 13, line 460).
Comment 7: I feel that the discussion misses an opportunity to explore why stigma may not differ by diagnostic label. Attribution theory or essentialism literature could be used to speculate further.
Response: We appreciate the reviewer’s suggestion and have integrated attribution theory into both the introduction (page 3, line 100) and the discussion (page 11, line 344).
Reviewer 2 Report
Comments and Suggestions for Authors
Dear authors, I have reviewed the manuscript titled “What is the Effect of Attributing Disordered Eating Behaviours to Food Addiction Versus Binge Eating Disorder? An Experimental Study Comparing the Impact on Weight-Based and Mental Illness Stigma. Overall, the study is well-structured, but some modifications are needed.
- I would suggest shortening the title for conciseness while retaining key elements. Lines 60-64: This paragraph is too long. Please simplify it.
- Line 177: Please remove the space before “assess”.
- Please add numbering for subparagraphs (e.g., 2.1. Participants, 2.2. Procedure, etc).
- Is there any protocol number for this study? If yes, please add it.
- The discussion is somewhat repetitive and could be more concise.
- Lines 387-390: Please add a reference for this paragraph.
- Line 416: The reference “(Mond et al., 2008)” does not meet the MDPI’s format. Please modify it.
- Line 261: Please remove the space before “vignette” and the extra dot at the end of the title.
- Please use consistent terminology (“Food Addiction” vs “FA”) and formatting for abbreviations across tables and text.
Author Response
Reviewer #2 comments:
Comment 1: I would suggest shortening the title for conciseness while retaining key elements.
Response: Though we appreciate the suggestion to shorten the title for conciseness, we believe that it is appropriately descriptive and have opted to retain it.
Comment 2: Lines 60-64: This paragraph is too long. Please simplify it.
Response: We have separated this paragraph into two separate sections (page 2, line 61).
Comment 3: Line 177: Please remove the space before “assess”.
Response: We have made this revision (page 4, line 186).
Comment 4: Please add numbering for subparagraphs (e.g., 2.1. Participants, 2.2. Procedure, etc).
Response: We appreciate this feedback as numbering the subparagraphs will make it easier for readers to understand. We have implemented this feedback into the paper.
Comment 5: Is there any protocol number for this study? If yes, please add it.
Response: Assuming the reviewer is referring to the protocol number for ethical approval, this number is included in the manuscript (page 3, line 140).
Comment 6 : The discussion is somewhat repetitive and could be more concise.
Response: We appreciate this feedback. We have reduced repetition throughout the discussion.
Comment 7: Lines 387-390: Please add a reference for this paragraph.
Response: We thank the reviewer for this suggestion. We included a reference; however, it did not adhere to the MDPI format. We have corrected the reference (page 12, line 420).
Comment 8: Line 416: The reference “(Mond et al., 2008)” does not meet the MDPI’s format. Please modify it.
Response: We thank the reviewer for identifying this error. We have changed the format of the citation to meet the MDPI format (page 12, line 420).
Comment 9: Line 261: Please remove the space before “vignette” and the extra dot at the end of the title.
Response: We have made these revisions to the manuscript (page 8, line 280).
Comment 10: Please use consistent terminology (“Food Addiction” vs “FA”) and formatting for abbreviations across tables and text.
Response: Thank you for bringing this inconsistency to our attention. We have now ensured that the acronym is used consistently throughout the body of the manuscript and in the tables.
Reviewer 3 Report
Comments and Suggestions for Authors
The paper “What is the Effect of Attributing Disordered Eating Behaviours to Food Addiction Versus Binge Eating Disorder? An Experimental Study Comparing the Impact on Weight-Based and Mental Illness Stigma” focuses on the differences in stigma between food addiction (FA) and binge eating disorder (BED), and explores the impact of diagnostic labels on weight and mental illness stigma through a vignette experimental design. The topic is of significant theoretical and practical importance. The research method is relatively standardized, and the data analysis is relatively comprehensive. The following are the specific review comments:
Comments:
Q1. It is suggested that the limitations of the sample be clearly stated in the discussion section and that more diverse samples be included in future research.
Q2. It is suggested that the interaction between gender perception and diagnostic labels be further explored during data analysis, or that the gender of the vignette characters be explicitly manipulated in future studies to eliminate this confounding factor.
Q3. The references are outdated. It is suggested that the latest research be supplemented to enhance the timeliness of the literature review.
Q4. The conclusion should clearly define the application scope of the research and emphasize the need for further validation through clinical samples or longitudinal studies.
Author Response
Reviewer #3 comments:
Overall comment: The paper “What is the Effect of Attributing Disordered Eating Behaviours to Food Addiction Versus Binge Eating Disorder? An Experimental Study Comparing the Impact on Weight-Based and Mental Illness Stigma” focuses on the differences in stigma between food addiction (FA) and binge eating disorder (BED), and explores the impact of diagnostic labels on weight and mental illness stigma through a vignette experimental design. The topic is of significant theoretical and practical importance. The research method is relatively standardized, and the data analysis is relatively comprehensive.
Response: We thank the reviewer for their positive feedback and appreciate their individual comments to help strengthen our manuscript.
Comment 1: It is suggested that the limitations of the sample be clearly stated in the discussion section and that more diverse samples be included in future research.
Response: In response to this comment and to comment #5 by Reviewer #1 above, we have added limitations of the participant sample and recommendations for future research in the revised manuscript (page 13, line 456).
Comment 2: It is suggested that the interaction between gender perception and diagnostic labels be further explored during data analysis, or that the gender of the vignette characters be explicitly manipulated in future studies to eliminate this confounding factor.
Response: We thank the reviewer for their thoughtful suggestion. We have recommended that future research replicate and extend the study by experimentally manipulating the gender of the vignette character to examine the causal effect of patient gender on stigma (page 13, line 463).
Comment 3: The references are outdated. It is suggested that the latest research be supplemented to enhance the timeliness of the literature review.
Response: Much of the research cited in the manuscript has been published within the past 5-10 years. We also cited some older publications if they were seminal papers in the field (e.g., publications by Weiner and colleagues on attribution theory; highly cited publications by several research teams on the conceptualization, overlap, and distinction between food addiction and binge eating disorder), relevant review papers, or experimental studies implementing methodologies that were similar to the present study (e.g., using vignettes to examine stigma).
Comment 4: The conclusion should clearly define the application scope of the research and emphasize the need for further validation through clinical samples or longitudinal studies.
Response: The purpose of the present study was to examine whether attributing disordered eating behaviours to food addiction or binge eating disorder has implications for weight-based or mental illness stigma in a non-clinical sample rather than a clinical sample. The reason for this decision is that we were interested in examining whether providing a causal explanation for a person’s eating behaviours and body weight might impact external stigma towards those who experience loss of control over their eating. We appreciate the reviewer’s comment and agree that it would be interesting to replicate the study in a clinical sample of people experiencing food addiction or binge eating disorder; however, this is beyond the scope of the objectives of the current study which focuses on public perceptions of food addiction and binge eating disorder.
Round 2
Reviewer 1 Report
Comments and Suggestions for Authors
The authors have sufficiently addressed my comments